# Intelligent Traffic Management in Next-Generation Networks

Ons Aouedi *, Kandaraj Piamrat * and Benoît Parrein

LS2N—Laboratoire des Sciences du Numérique de Nantes, Université de Nantes, 44000 Nantes, France;
benoit.parrein@ls2n.fr
* Correspondence: ons.aouedi@ls2n.fr (O.A.); kandaraj.piamrat@ls2n.fr (K.P.)

**Abstract:** The recent development of smart devices has lead to an explosion in data generation and heterogeneity. Hence, current networks should evolve to become more intelligent, efficient, and most importantly, scalable in order to deal with the evolution of network traffic. In recent years, network softwarization has drawn significant attention from both industry and academia, as it is essential for the flexible control of networks. At the same time, machine learning (ML) and especially deep learning (DL) methods have also been deployed to solve complex problems without explicit programming. These methods can model and learn network traffic behavior using training data/environments. The research community has advocated the application of ML/DL in softwarized environments for network traffic management, including traffic classification, prediction, and anomaly detection. In this paper, we survey the state of the art on these topics. We start by presenting a comprehensive background beginning from conventional ML algorithms and DL and follow this with a focus on different dimensionality reduction techniques. Afterward, we present the study of ML/DL applications in sofwarized environments. Finally, we highlight the issues and challenges that should be considered.

**Keywords:** machine learning; deep learning; networking; software-defined networking; resource management; feature selection; feature extraction; deep neural network; network traffic

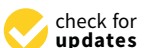



## 1. Introduction

According to the latest Cisco forecast, by 2022, the number of devices connected to mobile networks will largely exceed the world's population, reaching 12.3 billion [1]. This huge number of smart devices has made the Internet widely used and has, accordingly, triggered a surge in traffic and applications. This, in turn, has made network architecture highly resource-hungry and can create significant challenges for network operators. Therefore, it is not possible for network administrators to manage the network manually by processing a large amount of data accurately in a reasonable response time [2]. Consequently, this requires ultra-efficient, fast, and autonomous resource management approaches. In this context, machine learning (ML) offers these benefits, since it can find a useful pattern from data in a reasonable period of time and eventually allows the network operator to analyze the traffic and gain knowledge out of it. Network traffic analysis includes network traffic classification, network traffic prediction using time-series data, and intrusion detection systems. Understanding network behavior can improve performance, security, quality of service (QoS) and help avoid violation of SLA (service-level agreements). For example, load traffic prediction plays an important role in network management, such as short- and long-term resource allocation (e.g., bandwidth), anomaly detection, and traffic routing.

To achieve the intelligent management of networks and services in the software-defined networking (SDN) environment, ML methods can be used. A centralized SDN controller has a global network view which facilitates the collection of the network traffic. Several researchers argue that, with the introduction of SDN, there is a high potential for collecting data from forwarding devices, which need to be handled by ML due to their complexity [2,3]. Mestres et al. [3] presented a new paradigm that combines ML and SDN

which is called knowledge-defined networking (KDN). This architecture consists of knowledge, controller, and data planes in order to produce an automatic and intelligent process of network control. The adoption of ML with SDN for network management is an interesting area that requires further exploration (i.e., network slicing, traffic prediction). In addition, the availability of low-cost storage from the Cloud, edge computing, and the computer's evolution provide the high processing capabilities needed for training/testing ML and DL algorithms. Therefore, the combination of machine learning with SDN creates high-quality evidence in network traffic management. In this context, we provide a thorough overview of the research works in this area to gain deep insight into the significant role of ML/DL algorithms in the SDN environment.

### 1.1. Related Work

Driven by the recent advances in ML/DL in network traffic management, many studies have been conducted to survey the works on this domain. For example, Xie et al. [4] have provided a comprehensive survey of ML techniques applied to SDN. Different types of ML/DL algorithms and their applications in the SDN environment are presented. Nevertheless, they did not review the traffic prediction application. Additionally, the paper lacks several recent approaches as well as a dataset in the literature review.

Latah and Toker [5] briefly reviewed the application of AI with SDN. Then, in [6], they presented an extensive overview of AI techniques that have been used in the context of SDN. Additionally, Zhao et al. [7] surveyed ML/DL algorithms and their applications in the SDN environment. However, these papers categorized the reviewed contribution based on the ML model (e.g., supervised, unsupervised) and not by the use case.

A review on the approaches of traffic classification in software-defined wireless sensor networks (SDWSN) using ML has been presented in [8]. Mohammed et al. [9] briefly surveyed traffic classification and traffic prediction using ML in SDN. Moreover, Boutaba et al. [10] proposed a survey of the ML-based methods applied to fundamental problems in networking. In addition, a survey of SDN-based network intrusion detection systems using ML/DL approaches is presented in [11]. Nguyen et al. [12] proposed a survey on different security challenges related to the application of ML/DL in an SDN environment. As the main difference, these papers did not cover all the network traffic management applications in one paper. Furthermore, the research challenges in our paper are significantly different than those papers.

Table 1 summarizes existing survey articles with similar topics as ours. We categorize these papers into (i) overviews of deep learning, (ii) overviews of other machine learning algorithms, (iii) reviews of dimensionality reduction methods, and (iv) reviews of works at the intersection between DL/ML and SDN. The previous surveys have focused on ML/DL approaches in the SDN environment. However, they did not cover the traffic analysis applications like traffic classification, prediction, and anomaly detection. Additionally, they are lacking some new approaches. such as federated learning. In addition, while DL has the ability to learn features automatically, conventional ML algorithms try to extract knowledge from a set of features or attributes, but they do not have the capability to extract these features automatically. Therefore, choosing the appropriate features to be used with ML models is a predominant challenge. In this context, one important objective of this paper is to provide an overview ranging from dimensionality reduction and machine learning techniques to deep learning used in the SDN environment. Then, we explore the network traffic management/analysis that has benefited from ML/DL algorithms and SDN architecture.

**Table 1.** Summary of existing surveys related to ML/DL, dimensionality reduction (DR), and SDN.

| Ref. | Topic | Scope | | | |
|---|---|---|---|---|---|
| | | DL | Other ML Models | DR | SDN |
| [13] | A survey of deep learning and its applications. | ✓ | | | |
| [11] | A brief survey of SDN-based network intrusion detection system using machine and deep learning approaches. | ✓ | ✓ | | ✓ |
| [14] | A tutorial on deep learning. | ✓ | | | |
| [6] | An overview of AI techniques in the context of SDN. | ✓ | ✓ | | ✓ |
| [4] | A survey of ML/DL applications in SDN. | ✓ | ✓ | | ✓ |
| [15] | A brief survey of feature selection/extraction methods. | | | ✓ | |
| [16] | A survey of the procedure of feature selection and its application. | | | ✓ | |
| [17] | A survey of the software-defined wireless sensor networks. | | | | ✓ |
| [10] | A survey of the ML-based model applied to fundamental problems in networking | | ✓ | | |
| Our paper | A survey of DR, ML and DL in the SDN environment. | ✓ | ✓ | ✓ | ✓ |

## 1.2. Contributions

This paper is intended for researchers and developers who want to build intelligent traffic management as well as learning solutions in the SDN realm using the emerging ML/DL approaches. This paper discusses the role of the dimensionality reduction technique for conventional ML models as well as DL algorithms in SDN and how these concepts improve network traffic management. We cannot claim to have looked at each and every work under this combination (SDN and ML/DL), but we have covered the major approaches presented in the literature. In brief, the contributions of this survey can be summarized as follows:

- We provide a comprehensive view of the state-of-the-art machine/deep learning algorithms used in the SDN;
- We present the benefits of feature selection/extraction with conventional ML algorithms;
- We review the approaches and technologies for deploying DL/ML on SDN ranging from traffic classification to traffic prediction and intrusion detection systems;
- We highlight the problems and challenges encountered when using DL/ML.

## 1.3. Paper Organization

As illustrated in Figure 1, the rest of this paper is organized as follows. In Section 2, we present the main ML and DL-based models that have been used in the SDN environment and their strengths/weaknesses. Then, we present a comparison and summary table for the reviewed ML/DL algorithms. Section 2.3 presents a review of dimensionality reduction techniques, their advantages/disadvantages, and then we analyze how these techniques can be used to achieve the high performance of learning algorithms that ultimately improve the performance of the model. The main purpose of Section 3 is to provide a comprehensive survey of ML/DL applications with SDN for network traffic management/analysis. This survey also aims to highlight the list of challenges and issues of using ML/DL over SDN as presented in Section 4. Finally, Section 5 outlines the conclusions of this paper. For better comprehension, we summarize definitions of the abbreviations that will be used in this paper in Table 2.

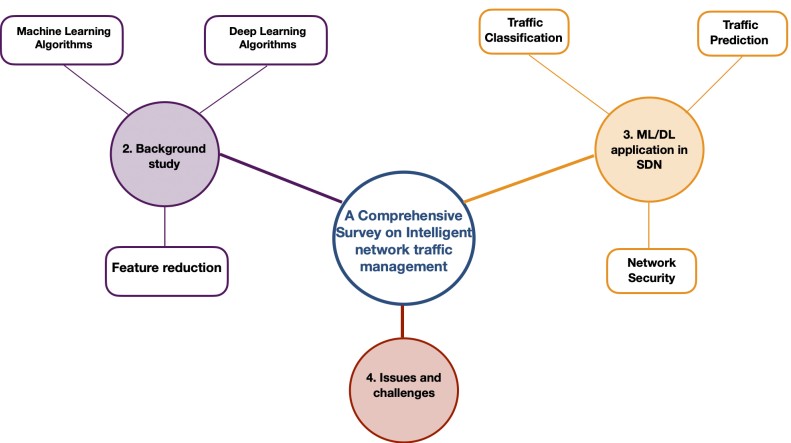

**Figure 1.** Conceptual map of survey.

**Table 2.** List of abbreviations.

| Acronym | Definition | Acronym | Definition |
|---------|-----------|---------|-----------|
| SDN | Software-defined networking | MLP | Multi-layer perceptron |
| SDWSN | Software-defined wireless sensor networks | CNN | Convolutional neural network |
| ML | Machine learning | LSTM | Long short-term memory |
| DL | Deep learning | MVNO | Mobile virtual network operator |
| KDN | Knowledge-defined networking | AE | Autodncoder |
| AI | Artificial intelligence | DR | Dimensionality reduction |
| QoS | Quality of service | QoE | Quality of experience |
| ANN | Artificial neural network | RL | Reinforcement learning |
| ONF | Open network foundation | OF | OpenFlow |
| NFV | Network function virtualisation | FL | Federated learning |
| DBN | Deep belief network | DRL | Deep Reinforcement Learning |
| GRU | Gated recurrent units | NGMN | Next Generation Mobile Networks |

## 2. Machine Learning

ML [18–20] is a branch of artificial intelligence (AI). Arthur Samuel was among the first researchers who applied machine learning to teach a computer to improve playing checkers based on training with a human counterpart in 1959. The increase in the amount of data made the ML algorithms popular in the 1990s. Nowadays, it is an emerging area that attracts the attention of academia and industry; Google, Apple, Facebook, Netflix, and Amazon are investing in ML and using it in their products [21]. Its effectiveness has been validated in different application scenarios, i.e., healthcare, computer vision, autonomous driving, recommendation systems, network traffic management, etc. It addresses the question of how to build a computer system that improves automatically through experience [19]. ML algorithms try to automate the process of knowledge extraction from training data to make predictions of unseen data. For example, historical traffic data are used to improve traffic classification and reduce congestion. In other words, ML models generally proceed in two phases: (1) in the training process, a training set is used to construct a model; (2) this model is then applied to predict or classify the unseen data. Therefore, the main idea of ML is to generalize beyond the examples in the training set and can be thought of as "programming by example". Hence, it is an intelligent method used to automatically improve performance through experience. Mitchell [20] defined ML by saying, *"A computer program is said to learn from experience E with respect to some class of tasks T and performance measure P, if its performance at tasks in T, as measured by P, improves with experience E"*.

The adoption of ML and DL approaches increased thanks to the computer's evolution, which has provided us with processing capabilities needed for training/testing their models, the availability of large datasets, and the cost-effectiveness of storage capabilities. In addition, the availability of open-source libraries and frameworks causes rapid diffusion within the research community and in our daily life. ML/DL algorithms can

be implemented in various platforms and languages such as Weka (does not include DL algorithms), R, and Python. Python is one of the most popular environments for ML/DL as it provides various libraries to developers. Additionally, its simplicity helps developers to save more time, as they only require concentrating on solving the ML/DL problems rather than focusing on the technicality of language. The most commonly used tool for classical machine learning implementation is scikit-learn [22]. Scikit-learn is an open-source ML library (e.g., Matplotlib, Pandas) and is not specific to DL. It covers all data mining steps from preprocessing to classification, regression, and clustering.

Within the field of machine learning, a broad distinction could be made between supervised, unsupervised, semi-supervised, and reinforcement learning (RL). The strengths and weaknesses of these learning approaches are presented in Table 3.

- **Supervised learning**

This type of learning process is the most commonly used. It operates when an object needs to be assigned into a predefined class based on several observed features related to that object [23]. Therefore, supervised learning tries to find model parameters that best predict the data based on a loss function $L(y, \hat{y})$. Here, $y$ is our output variable, and $\hat{y}$ represents the output of the model obtained by feeding a data point $x$ (input data) to the function $F$ that represents the model. Classification and regression are examples of supervised learning: in classification, the outputs take discrete values; this is one of the most widely used methods. In regression, a learning function maps the data into a real-value variable (the outputs are continuous values). Binary, multi-class, and multi-labeled classification are the three approaches of classification [24]. In binary classification, only two possible classes, for example, classify the traffic as "attack" or "normal". Multi-class classification implies that the input can be classified into only one class within a pool of classes, such as classifying the traffic as "Chat", "Streaming", and "Game". Multi-labeled classification allows the classification of an input sample into more than one class in the pool of classes, like classifying the traffic as "Skype" and the traffic type as "Video". Classification examples include decision tree, support vector machine (SVM), and neural networks, while for regression, we have support vector regression (SVR), linear regression, and neural networks. In addition, an ensemble of different learning algorithms was developed to produce a superior accuracy than any single algorithm like random forest and boosting methods [25];

- **Unsupervised learning**

With the increasing size and complexity of data, unsupervised learning can be one promising solution. It has the ability to find a relationship between the instances without having any prior knowledge of target features. Specifically, unsupervised learning examines the similarity between the instances in order to categorize them into distinctive clusters. The instances within the same cluster have a greater similarity as compared to the instances in other clusters [26];

- **Semi-supervised learning**

As the name implies, semi-supervised learning combines both supervised and unsupervised learning to get benefits from both approaches. It attempts to use unlabeled data as well as labeled data to train the model, contrary to supervised learning (data all labeled) and unsupervised learning (data all unlabeled). As labeling data (e.g., network traffic) is difficult, requires human effort, and is time-consuming to obtain, especially for traffic classification and attack detection, semi-supervised learning tries to minimize these problems, as it uses a few labeled examples with a large collection of unlabeled data [27]. It is an appropriate method when large amounts of labeled data are unavailable. That is why, in the last few years, there has been a growing interest in semi-supervised learning in the scientific community, especially for traffic classification [28];

- **Reinforcement learning (RL)**

The main idea of reinforcement learning was inspired by biological learning systems. It is different from supervised and unsupervised learning; instead of trying to find a pattern or learning from a training set of labeled data, the only source of data for RL is the feedback a software agent receives from its environment after executing an action [29]. That is why it is considered as a fourth machine learning approach alongside supervised, unsupervised, and semi-supervised learning. In addition, RL is defined by the provision of the training data by the environment. In other words, it is a technique that allows an agent to learn its behavior by interacting with its environment. Three important elements construct this learning approach, namely observations, reward, and action. Therefore, the software agent makes observations and executes actions within an environment, and receives rewards in return. The agent's job is to maximize cumulative reward. The most well-known reinforcement learning technique is Q-learning [30], and it is widely used for routing [31,32]. Moreover, deep learning has been used to improve the performance of RL algorithms (i.e., allows reinforcement learning to be applied to larger problems). Therefore, the combination of DL and RL gives the so-called DRL. DRL began in 2013 with Google Deep Mind [33]. A good survey that presents RL and DRL approaches is available in [34] for more details.

**Table 3.** Comparison of ML approaches.

| Method | Strengths | Weaknesses |
|---|---|---|
| Supervised learning | Low computational cost, fast, scalable | Requires data labeling and data training, behaves poorly with highly imbalanced data |
| Unsupervised learning | Requires only the data samples, can detect unknown patterns, generates labeling data | Cannot give precise information |
| Semi-supervised learning | Learns from both labeled and unlabeled data | May lead to worse performance when we choose the wrong rate of unlabeled data |
| Reinforcement learning | Can be used to solve complex problems, efficient when the only way to collect information about the environment is to interact with it | Slow in terms of convergence, needs a lot of data and a lot of computation |

In Section 2.1, a synthesis of DL architecture and models is presented, followed by an overview of conventional ML models as well as feature reduction techniques. We only focus on the algorithms that have been used in the SDN environment and that are presented in Section 3.

*2.1. Deep Learning*

Deep learning (DL), also known as deep neural networks (DNN) represents one of the most active areas of AI research [13]. It is a branch of ML that evolved from neural network (NNs) which enables an algorithm to make predictions or classifications based on large datasets without being explicitly programmed. In many domains, DL algorithms are able to exceed human accuracy. Using conventional ML-based models requires some feature engineering tasks like feature selection [35], which is not the case with DL models, as these can hierarchically extract knowledge automatically from raw data by stacked layers [23].

The major benefits of DL over conventional ML models are its superior performance for large datasets [36] and the integration of feature learning and model training in one architecture, as illustrated in Figure 2. Therefore, it helps to avoid human intervention and time-wasting as maximum as possible. In the literature, DL has also been referred to as deep structured learning, hierarchical learning, and deep feature learning [37]. This deep architecture enables DL to perform better than conventional ML algorithms; accordingly, DL can learn highly complicated patterns. It uses supervised and unsupervised learning to learn high-level features for the tasks of classification and pattern recognition. Several DL algorithms are presented in Table 4. The growing popularity of DL inspired several compa-

nies, and open-source initiatives have developed powerful DL libraries and frameworks that can be used to avoid building models from scratch [38]. Many libraries have appeared, and we summarize the most popular ones in Table 5.

DL is capable of learning high-level features better than shallow neural networks. Shallow ANN contains very few hidden layers (i.e., with one hidden layer) while DL contains many more layers (deep). Each hidden layer comprises a set of learning units called neurons. These neurons are organized in successive layers, and every layer takes as input the output produced by the previous layer, except for the first layer, which consumes the input.

**Table 4.** Summary of different deep learning models used in SDN.

| Method | Learning Model | Description | Strengths | Weaknesses |
|---|---|---|---|---|
| MLP | Supervised, unsupervised | MLP is a simple artificial neural network (ANN) which consists of three layers. The first layer is the input layer. The second layer is used to extract features from the input. The last layer is the output layer. The layers are composed of several neurons. | Easy to implement. | Modest performance, slow convergence, occupies a large amount of memory. |
| AE | Unsupervised | Autoencoder consists of three parts: (i) encoder, (ii) code, and (iii) decoder blocks. The encoder converts the input features into an abstraction, known as a code. Using the code, the decoder tries to reconstruct the input features. It uses some non-linear hidden layers to reduce the input features. | Works with big and unlabeled data, suitable for feature extraction and used in place of manual engineering. | The quality of features depends model architecture and its hyperparameters, hard to find the code layer size. |
| CNN | Supervised, Unsupervised | CNN is a class of DL which consists of a number of convolution and pooling (subsampling) layers followed by a fully connected layers. Pooling and convolution layers are used to reduce the dimensions of features and find useful patterns. Next, fully connected layers are used for classification. It is widely used for image recognition applications. | Weight sharing, extracts relevant features, high competitive performance. | High computational cost, requires large training dataset and high number of hyperparameter tuning to achieve optimal features. |
| LSTM | Supervised | LSTM is an extension of recurrent neural network (RNNs) and was created as the solution to short-term memory. It has internal mechanisms called gates (forget gate, input gate, and output gate) that can learn which data in a sequence is important to keep or throw away. Therefore, it chooses which information is relevant to remember or forget during sequence processing. | Good for sequential information, works well with long sequences. | High model complexity, high computational cost. |
| GRU | Supervised | GRU was proposed in 2014. It is similar to LSTM but has fewer parameters. It works well with sequential data, as does LSTM. However, unlike LSTM, GRU has two gates, which are the update gate and reset gate; hence, it is less complex. | Computationally more efficient than LSTM. | Less efficient in accuracy than LSTM. |
| DRL | Reinforcement | DRL takes advantage of both DL and RL to be applied to larger problems. In others word, DL enables RL to scale to decision-making problems that were previously intractable. | Scalable (i.e., can learn a more complex environment). | Slow in terms of training. |
| DBN | Unsupervised, supervised | DBN is stacked by several restricted Boltzmann machines. It takes advantage of the greedy learning process to initialize the model parameters and then fine-tunes the whole model using the label. | Training is unsupervised, which removes the necessity of labelling data for training or properly initializing the network, which can avoid the local optima, extracting robust features. | High computational cost. |

**Table 5.** Summary of popular deep learning frameworks [38,39].

| DL Frameworks | Creator | Available Interface | Popularity | Released |
|---|---|---|---|---|
| Tensorflow | Google Brain Team | C++, Go, Java, JavaScript, Python, Swift | High | 2015 |
| Caffe2 | Facebook AI research | C++, Python | Low | 2017 |
| Deeplearning4j | Skymind | Java, Scala | Low | 2014 |
| MXNet | Apache Software Foundation | C++, Python, Julia, Matlab, JavaScript, Go, R, Scala, Perl | Medium | 2015 |
| Theano | University of Montreal | Python | Medium | 2017 |
| CNTK | Microsoft Research | C++, C#, Python | Low | 2016 |
| PyTorch | Facebook AI research | C++, Python | High | 2016 |
| Keras (higher level library for TensorFlow, CNTK, Theano, etc.) | François Chollet | Python | High | 2015 |

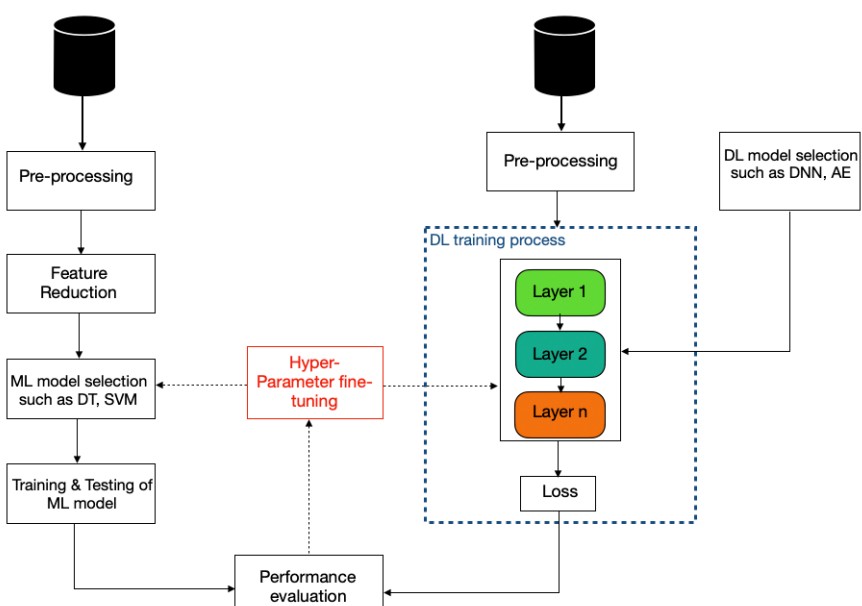

**Figure 2.** Conventional ML and DL.

More specifically, the neuron receives a vector $x$ as input and uses it to compute an output signal, which is transferred to other neurons. It is parameterized through $\{W, b\}$, where $W$ is a weight vector, $b$ is the bias, and $f$ is referred to as an activation function. In modern networks, activation functions are referred to as non-linear functions. This non-linearity enables the DL model to learn complex patterns and avoid constraints associated with linear functions. The choice of activation function affects network training time [40]. The most frequently used activation functions are the ReLU (rectified linear unit), sigmoid, and hyperbolic tangent functions (TanH). In fact, networks with ReLu show better convergence performance than sigmoid and tanh [40]. For more details of ANN, readers can refer to the book written by Haykin [41].

*2.2. Conventional Machine Learning Models*

Although DL-based models perform well for complex applications, there still remain several challenges. For example, the number of their hyperparameters grows exponentially with the depth of the model. Additionally, finding suitable DL architecture (i.e., number of hidden layers) and identifying optimal hyperparameters (i.e., learning rate, loss function, etc.) are difficult tasks. Furthermore, DL does not perform well when the data volume is small because it needs a large amount of data to find some patterns from the data. In such a context, conventional ML models (e.g., decision tree) can provide better results with a minimal fine-tuning process of the hyperparameters. In addition, compared to the

conventional ML models, DL has a higher computational cost, which makes it hard to be used on machines with a simple CPU.

All this makes the conventional ML models still used. They can be broadly divided into two categories: (i) simple (i.e., single) machine models and (ii) ensemble models. The commonly used simple models include, for example, SVM, DT, and KNN, whereas ensemble models aim to combine heterogeneous or homogeneous simple models in order to obtain a model that outperforms every one of them and overcomes their limitations [42,43]. The ensemble methodology imitates our nature to seek several opinions before making a decision [44]. Bagging, boosting, and stacked generalization (or simply stacking) are the most popular ensemble models. Similar to single models, there are no best ensemble methods, as some ensemble methods work better than others in certain conditions (e.g., data, features). Several conventional ML methods used with SDN are presented in Table 6.

However, the performance of the conventional ML models depends on the amount and the quality of features. Moreover, extracting a large number of features from the coming flow in the SDN environment is time-consuming and, in turn, can decrease the QoS of the system [45]. To solve these issues, effective pre-processing mechanisms can be applied to filter out the unrelated/noisy data that can reduce the dimensionality of the data.

**Table 6.** Comparison of classical ML approaches used in SDN.

| Method | Description | Strengths | Weaknesses |
|---|---|---|---|
| Decision tree (DT) | Decision tree is a tree-like structure, where every leaf (terminal) corresponds to a class label and each internal node corresponds to an attribute. The node at the top of the tree is called the root node. Tree splitting uses Gini Index or Information Gain methods [46]. | Simple to understand and interpret, requires little data preparation, handles many types of data (numeric, categorical), easily processesdata with high dimension | Generates a complex tree with numeric data, requires large storage |
| Random forest (RF) | Random forest was developed nearly 20 years ago [47] and is one of the most popular supervised machine learning algorithms that is capable to be used for regression and classification. As their name would suggest, random forests are constructed from decision trees. It uses the bagging method, which enhances the performance. | Efficient against over-fitting | Requires a large training dataset, impractical for real-time applications |
| Support vector machine (SVM) | SVM is a powerful classification algorithm which can be used for both regression and classification problems. However, it is mostly used as classification technique. It was initially developed for binary classification, but it could be efficiently extended to multiclass problems. It can be a linear and non-linear classifier by creating a splitting hyperplane in original input space to separate the data points. | Scalable, handles complex data | Computationally expensive, there is no theorem to select the right kernel function |
| K-nearest neighbour (KNN) | KNN is a supervised model reason with the underlying principal "Tell me who are your friends, I will tell you who are you" [48]. It classifies new instances using the information provided by the K nearest neighbors, so that the assigned class will be the most common among them (majority vote). Additionally, as it does not build a model and no work is done until an unlabeled data pattern arrives, it is thus considered as a lazy approach. | Easy to implement, has good performance with simple problems, non-expert users can use it efficiently | Requires large storage space, determining the optimal value of K is time consuming, K values varies depending on the dataset, testing is slow, and when the training dataset is large, it is not suitable for real-time classification |
| K-means | K-means is a well-known unsupervised model. It can partition the data into K clusters based on a similarity measure, and the observations belonging to the same cluster have high similarity as compared to those of other clusters. | Fast, simple and less complex | Requires a number of cluster in advance, cannot handle the outliers |
| Boosting algorithms | The main idea of boosting is to improve the performance of any model, even weak learners (i.e., XGBoost, AdaBoost, etc). The base model generates a weak prediction rule, and after several rounds, the boosting algorithms improve the prediction performance by combining the weak rules into a single prediction rule [49]. | High accuracy, efficient against under-fitting | Computationally expensive, hard to find the optimal parameters |

*2.3. Feature Reduction*

The success of conventional ML algorithms generally depends on the features to which they are applied. In other words, their performance depends not only on the parameters but also on the features, which are used to describe each object to the ML models [50]. In this section, we provide an overview of basic concepts as well as the motivation of the dimensionality reduction technique.

In general, the dataset can contain irrelevant (a feature that provides no useful information) and redundant (a feature with predictive ability covered by another) features, which cause an extra computational cost for both storage and processing in an SDN environment and decreases the model's performance [51,52]. For example, the time complexity of many ML algorithms is polynomial in the number of dimensions; for example, the time complexity of logistic regression is $O(mn^2 + n^3)$, where $n$ represents the number of features and $m$ represents the number of instances [53]. These challenges are referred to as the *curse of dimensionality*, which is one of the most important challenges in ML. The curse of dimensionality was introduced by Bellman in 1961 [18].

In addition, each model is only as good as the given features. Additionally, the model accuracy depends not only on the quality of input data but also on the amount features. The model may perform poorly and can overfit if the number of features is larger than the number of observations [54]. Moreover, the low-latency applications (e.g., anomaly detection) require fast analysis, especially in the context of the fifth-generation (5G) networks, and feature reduction can help to leverage heavy analysis at network levels. All these make feature reduction difficult tasks not only because of the higher size of the initial dataset but because it needs to meet two challenges, which are to maximize the learning capacity and to reduce the computational cost and delay by reducing the number of features.

In fact, there are many benefits of feature reduction techniques, such as (i) improving the performance of learning algorithms either in terms of learning speed or generalization capacity and (ii) reducing the storage requirement. It reduces the original data and retains as much information as possible that is valuable for the learning process. Therefore, in a dataset with a high number of features, the data reduction process is a must to produce accurate ML models in a reasonable time. It can be divided into feature selection (i.e feature elimination) and feature extraction, as shown in Figure 3. The main difference is that feature selection selects a subset of original features, while feature extraction creates new features from the original ones. Table 7 presents the advantages and disadvantages of each approach.

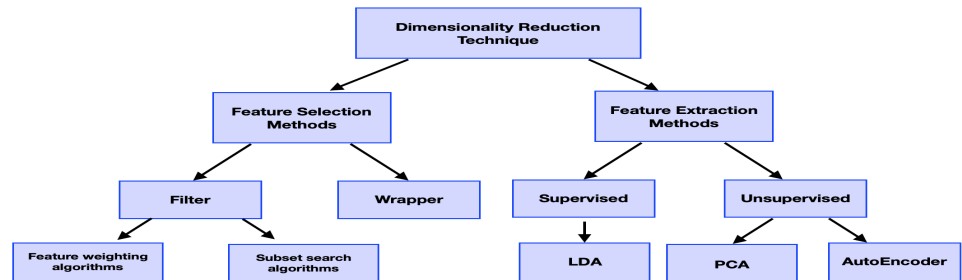

**Figure 3.** The classification of dimensionality reduction approaches.

Each approach can be used in isolation or in a combination way with the aim of improving performance, such as the accuracy, visualization, and comprehensibility of learned knowledge [55]. Rangarajan et al. [56] proposed bi-level dimensionality reduction methods that combine feature selection methods and feature extraction methods to improve classification performance. Then, they compared the performance of three-dimensionality reduction techniques with three datasets. Based on their results, these combinations improve the precision, recall, and F-measure of the classifier. Additionally, we can find feature construction, which is different from the feature reduction technique. It is one of the

preprocessing tasks that tries to increase the expressive power of initial features when the initial features set do not help us classify our data well enough. It is the process of creating additional features instead of reducing the features from the initial features set, beginning with $n$ features $f_1, f_2, f_3, \ldots, f_n$. After feature construction, we will have new $p$ features $f_{n+1}, f_{n+2}, f_{n+3}, \ldots, f_{n+p}$.

**Table 7.** Comparison of dimensionality reduction techniques [16,57].

| Method | Advantages | Disadvantages | Methods | Potential Application in SDN |
|---|---|---|---|---|
| Filter | Low computational cost, fast, scalable | Ignores the interaction with the classifier | CFS, IG, FCBF | QoS prediction [58], traffic classification [59,60]. |
| Wrapper | Competitive classification accuracy, interaction with the classifier | Slow, expensive for large feature space, risk of over-fitting | Forward/backward direction | Traffic classification [61], QoS prediction [58]. |
| Feature extraction | Reduces dimension without loss of information | No information about the original features | PCA, LDA, AE | Traffic classification [59,62,63]. |

### 2.3.1. Feature Selection

The feature selection technique has become an indispensable component of the conventional machine learning algorithms (e.g., SVM, decision tree, etc.) [64], and it is one of the simplest ways to reduce data size. It tries to pick a subset of features that "optimally" characterize the target variable. Feature selection is the process of selecting the best features in a given initial set of features that yield a better classification performance [65] and regression as well as finding clusters efficiently [66]. In this section, we will focus on feature selection for classification since there is much more work on feature selection for classification than for other ML tasks (e.g., clustering). The best subset contains the least number of dimensions that improve the accuracy the most. It helps us to identify the relevant features (contribute to the identification of the output) and discard the irrelevant ones from the dataset to perform a more focused and faster analysis. The feature selection process consists of four basic steps, which are:

1. *Subset generation* is a search procedure that generates candidate feature subsets for evaluation based on a search strategy (i.e., start with no feature or with all features);
2. *Evaluation of subset* tries to measure the discriminating ability of a feature or a subset to distinguish the target variables;
3. *Stopping criteria* determines when the feature selection process should stop (i.e., addition or deletion of any feature does not produce a better subset);
4. *Result validation* tries to test the validity of the selected features.

Many evaluation criteria have been proposed in several research works to determine the quality of the candidate subset of the features. Based on their dependency on ML algorithms, evaluation criteria can be categorized into two groups: independent and dependent criteria [16]. For independent measures, we have (1) information measures, (2) consistency measures, (3) correlation measures, (4) and distance measures; for dependent criteria, we have several measures, such as accuracy. Therefore, feature selection can be distinguished into two broad categories, which are *filters* and *wrappers*, as explained in the following.

**Filter Method**: finds the best feature set by using some independent criteria (i.e., information measures) before applying any classification algorithm. Due to the computational efficiency, the filter methods are used to select features from high-dimensional data sets. Additionally, it is categorized as a binary or continuous feature selection method depending on the data type. For example, information gain (IG) can handle both binary and nominal data, but the Pearson correlation coefficient can only handle continuous data. Moreover, it can be further categorized into two groups: feature-weighting algorithms and subset search algorithms, which evaluate the quality of the features individually or through feature subsets. Feature-weighting algorithms give weights to all the features individually and rank them based on their weights. These algorithms can be computationally inexpensive

and do not consider feature interactions, which may lead to selecting redundant features. This is why subset search algorithms have become popular. Subset search algorithms search through candidate feature subsets using certain evaluation measures (i.e., correlation) that capture the goodness of each subset [67];

**Wrapper Method**: requires a learning algorithm and uses its performance as the evaluation criterion (i.e., classifier accuracy). It calculates the estimated accuracy of a single learning algorithm through a search procedure in the space of possible features in order to find the best one. The search can be done with various strategies, such as forwarding direction (the search begins with an empty set and successively adds the most relevant features) and backward direction (the search starts with the full set and successively deletes less relevant features), which is also known as recursive feature elimination. Wrapper methods are also more computationally expensive than the filter methods and feature extraction methods, but they produce feature subsets with very competitive classification accuracy [25,52].

### 2.3.2. Feature Extraction

Feature extraction performs a transformation of the original variables to generate other features using the mapping function $F$ that preserves most of the relevant information. This transformation can be achieved by a linear or non-linear combination of original features. For example, for $n$ features $f_1, f_2, f_3, \ldots, f_n$, we extract a new feature set $f'_1, f'_2, f'_3, \ldots, f'_m$ where $m < n$ and $f'_i = F(f_1, f_2, f_3, \ldots, f_n)$. With the feature extraction technique, the feature space can often be decreased without losing a lot of information on the original attribute space. However, one of its limits is that the information about how the initial features contribute is often lost [52]. Moreover, it is difficult to find a relationship between the original features and the new features. Therefore, the analysis of the new features is almost impossible, since no physical meaning for the transformed features is obtained from feature extraction techniques. In this survey, we discuss three of the most frequently used methods.

*Principal Component Analysis (PCA)* is the oldest technique of multivariate analysis and was introduced by Karl Pearson in 1901. It is an unsupervised (it does not take into account target variable) and non-parametric technique that reduces the dimensionality of data from $f$ to $p$, where $p < f$, by transforming the initial feature space into a smaller space. However, PCA has some limitations, discussed below:

- It assumes that the relations between variables are linear;
- It depends on the scaling of the data (i.e., each variable is normalized to zero mean);
- We do not know how many PCs should be retained (the optimal number of principal components (PCs));
- It does not consider the correlation between target outputs and input features [23];

*Autoencoder (AE)* is an unsupervised learning model which seeks to reconstruct the input from the hidden layer [68]. During the process, the AE tries to minimize the reconstruction error by improving the quality of the learned feature. It is frequently used to learn discriminative features of original data. Hence, AE is potentially important for feature extraction, and many researchers use it to generate reduced feature sets (code). In many cases, deep autoencoder (DAE) outperforms conventional feature selection methods and PCA [69] since it consists of several layers with nonlinear activation functions to extract features intelligently;

*Linear discriminant analysis (LDA)* is a feature extraction method used as a preprocessing step for ML algorithms and was first proposed in [70]. It is similar to PCA, but LDA is a supervised method. In other words, it takes the target variable into account. It consists of three main steps, as mentioned in [71]. Firstly, it calculates the *between-class variance* using the distance between the means of the classes. Secondly, it calculates the *within-class variance* using the distance between the mean and the observations of each class. Finally, it constructs the lower dimensional space, which maximizes the between-class variance and minimizes the within-class variance.

*2.4. Federated Learning*

Federated learning (FL) is a decentralized learning approach used to ensure data privacy and decrease the exchange message between the server and clients [72]. As shown in Figure 4, the training process is divided into two steps: (i) local model training and (ii) global aggregation of updated parameters in the FL server. For more details, first, the central server specifies the hyperparameters of the global model and the training process. Then, it broadcasts the global model to several clients. Based on this model, each client trains the global model for a given number of epochs using its own data. Then, they send back the updated model to the FL server for the global aggregation. These steps are repeated until the global model is achieved for a selected number of rounds in order to achieve satisfactory performance. As a result, FL helps the final model to take advantage of the different clients without exchanging their data. Additionally, it may decrease the communication overhead by exchanging the model parameters instead of the clients' data.

Since the traffic data has been increased, FL can be a promising solution for traffic management. For example, FL helps to detect the attack without communicating the packet/flow to a central entity [73]. It keeps the traffic where it was generated. As a result, FL has started to attract the attention of researchers for intrusion detection [74] and prediction [75].

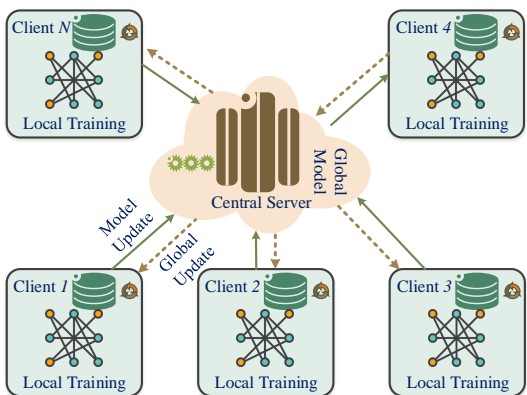

**Figure 4.** General federated learning architecture [73], local model training on the client-side and global aggregation of updated parameters in the FL server.

## 3. DL/ML Applications in Software-Defined Networking

Thanks to the main features of SDN, including the global network view, separation of the data plane from the control plane, and network programmability, SDN provides a new opportunity for ML/DL algorithms to be applied for network traffic management. In this section, we present the different application cases in the SDN environment using ML/DL algorithms in detail. According to the different application scenarios, we divide the existing application cases into three categories: traffic classification, traffic prediction, and intrusion detection systems.

*3.1. Traffic Classification*

The explosion in the number of smart devices has resulted in significant growth of network traffic as well as new traffic types. For example, mobile data traffic will be 77 exabytes per month, which is 7 times that in 2017 [1]. In this context, the purpose of traffic classification is to understand the type of traffic carried on the Internet [76]. It aims to identify the application's name (YouTube, Netflix, Twitter, etc.) or type of Internet traffic (streaming, web browsing, etc.). Therefore, traffic classification has significance in a variety of network-related activities, from security monitoring (e.g., detecting malicious attacks) and QoS provisioning to providing operators with useful forecasts for long-term and traffic management. For example, identifying the application from the traffic can help us to manage bandwidth resources [24]. Additionally,

network operators need to know what is flowing over their networks promptly so they can react quickly in support of their various business goals [77]. Thereby, traffic classification has evolved significantly over time. It tries to separate several applications based on their feature "profile". Because of its importance, several approaches have been developed over the years to accommodate the diverse and changing needs of different application scenarios. There are three major approaches to achieve application identification and classification: (i) port-based, (ii) payload-based (deep packet inspection), and (iii) ML–based [50]. Since this paper focuses on the application of ML/DL in softwarized networks, we will present only the approaches based on ML and DL models.

### 3.1.1. Existing Solutions

In this subsection, we study several works that try to achieve different traffic classification objectives. We have divided traffic classification solutions into two broad categories according to classification level: (i) coarse-grained and (ii) fine-grained. The coarse-grained level classifies the traffic based on the traffic type, while the fine-grained level aims to classify based on the exact application. At the end, we also highlight some other recent approaches.

- **Coarse-grained traffic classification**

With the intensive growth of online applications, it has become time-consuming and impractical to identify all the applications manually. Therefore, coarse-grained traffic classification aims to identify the QoS classes of traffic flows (e.g., streaming, web browsing, chat, etc.). Coarse-grained traffic classification is used to satisfy the QoS requirements of the network application at the same time.

Parsaei et al. [78] introduced a network traffic classification in the SDN environment where several variants of neural networks were applied to classify traffic flows into different QoS classes (instant messaging, video streaming, FTP, HTTP, and P2P). These classifiers provided 95.6%, 97%, 97%, and 97.6% in terms of accuracy. However, the authors did not mention information on the architecture of the different neural network algorithms.

Xiao et al. [79] proposed a classification framework based on SDN. It consists of two modules, which are the flow collector and spectral classifier. First, the flow collector module scans the flow tables from SDN controllers to collect traffic flows. Then, the spectral classifier module receives the flow from the flow collector module to create the clusters through spectral clustering. They fixed $k = 6$ as the number of flow classes (HTTP, SMTP, SSH, P2P, DNS, SSL2). Then, to evaluate their results, a comparative analysis with the K-means method in terms of accuracy, recall, and classification time was done. However, the authors did not mention information on the features and preprocessing methods used during their experiments.

Da Silva et al. [59] focused on the identification and selection of flow features to improve traffic classification in SDN. The authors introduced architecture to compute additional flow features and select the optimal subset of them to be applied in traffic classification. This architecture is composed of flow feature manager and flow feature selector. Flow feature manager is responsible for gathering information through the controller and extending this information to complex flow features. Then, the flow feature selector finds the optimal subset of flow features using principal component analysis (PCA) and genetic algorithm (GA), followed by the classifier (SVM) to evaluate this subset. Based on the experimental results, we can reveal some conclusions: this feature selection technique can improve traffic classification and the features generated in this paper can give good accuracy.

In the same direction, Zaki and Chin [80] compared four classifiers, C4.5, KNN, NB, and SVM. Those classifiers were trained with features selected by their proposed hybrid filter wrapper feature selection algorithm named (*FWFS*). *FWFS* evaluates the worth of the features with respect to the class, and the wrappers use an ML model to measure the feature subset. The experimental results demonstrate that the selected features through FWFS improve the performance of the classifiers.

Recently, the pattern of network traffic has become more complex, especially with the 5G network. This has made simple classifiers unable to provide an accurate model. To solve this issue, researchers in the field of networking started to use ensemble learning for network traffic classification. One of the main advantages of ensemble learning is its ability to allow the production of better predictive performance compared to a single model. In this context, Eom et al. [81] applied four ensemble algorithms (RF, gradient boosting machine, XGBoost, LightGBM) and analyzed their classification performance in terms of accuracy, precision, recall, FI-score, training time, and classification time. The experiment results demonstrate that the LightGBM model achieves the best classification performance.

Yang et al. [82] proposed a novel stacking ensemble classifier which combines seven models in two-tier architecture (Figure 5). Specifically, on the first tier, KNN, SVM, RF, AdaBoost, and gradient boosting machine are trained independently on the same training set. Then, on the second tier, XGboost uses the prediction of the first-tier classifiers in order to improve the final prediction results. The comparative analysis against the individual single classifier and voting ensemble demonstrates the effectiveness of the proposed ensemble.

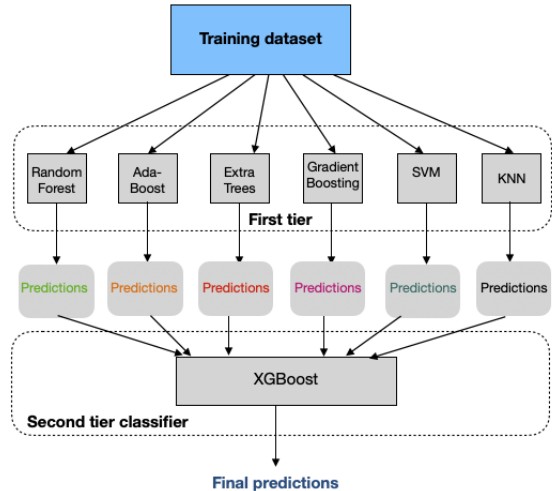

**Figure 5.** Flowchart for the proposed ensemble classifier discussed in Yang et al. [82].

As DL helps to eliminate the manual feature engineering task, Hu et al. [83] proposed a CNN-based deep learning method to address the SDN-based application awareness called *CDSA*. As shown in Figure 6, CDSA consists of three components, which are traffic collection, data pre-processing, and application awareness. To evaluate the performance of this framework, the open Moore dataset [84] has been used. This dataset was released by the University of Cambridge.

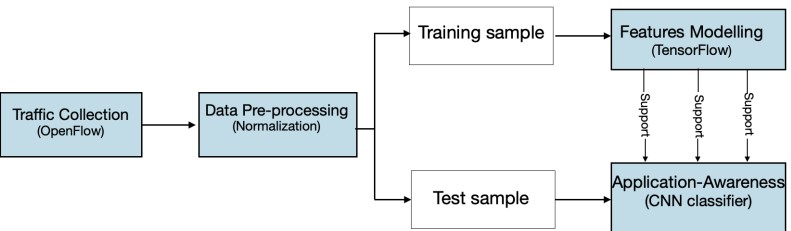

**Figure 6.** Framework of CDSA, which consists of traffic collection, data pre-processing and application awareness [83].

Malik et al. [85] introduced an application-aware classification framework for SDNs called *Deep-SDN*. The performance of Deep-SDN was evaluated on the Moore dataset [84], which is real-world traffic. The experimental results demonstrate that the Deep-SDN outperformed the DL model proposed in [63] by reporting 96% overall accuracy.

As new types of traffic emerge every day (and they are generally not labeled), this opens a new challenge to be handled. In fact, labeling data is often difficult and time-consuming [28]. Therefore, researchers started to reformulate traffic classification into semi-supervised learning, where both supervised learning (using labeled data) and unsupervised learning (no label data) were combined.

In such a context, a framework of a semi-supervised model for network traffic classification has been proposed in [61]. Specifically, DPI has been used to label a part of traffic flows of known applications. Each labeled application is categorized into four QoS classes (voice/video conference, interactive data, streaming, bulk data transfer). Then, this data is used by Laplacian SVM as a semi-supervised learning model in order to classify the traffic flows of unknown applications. Here, only "elephant" flows are used in the traffic classification engine. The proposed framework is fully located in the network controller and exceeds 90% accuracy. Additionally, the Laplacian SVM approach outperforms a previous semi-supervised approach based on the K-means classifier. Therefore, based on the experimental results presented in their paper, we can reveal some conclusions: by combining the DPI and ML method, traffic flows could be categorized into different QoS classes with acceptable accuracy. However, the authors did not present any detail about the complexity of the proposed framework.

Similar work is done by Yu et al. [86], where the authors proposed a novel SDN flow classification framework using DPI and semi-supervised ensemble learning. The significant difference is the ML algorithm used for the classification. Firstly, they used DPI to generate a partially labeled dataset with the QoS type of flow according to the application type. To train the classifier, they used the heteroid tri-training algorithm. This algorithm is a modified version of tri-training, which is a classic semi-supervised learning mechanism that uses three identical classifiers to enable iterative training. However, they used three heterogeneous classifiers (SVM, Bayes classifier, K-NN) instead of three identical classifiers to increase the difference among classifiers. To verify the performance of their framework, they used a real dataset and trained the classifier with different ratios of unlabeled application flows. However, the authors did not mention information about the feature selection method and the dataset used during the experiments.

In addition, Zhang et al. [63] proposed a DL-based model in the SDN-based environment to classify the traffic to one of several classes (Bulk, Database, Interactive, Mail, Services, WWW, P2P, Attack, Games, Multimedia). It consists of the stacked autoencoder (SAE) and softmax classifier. SAE was used for feature extraction and softmax was used as a supervised classifier. The experimental results show that the proposed model outperforms the SVM. However, the authors proposed only the framework without testing its performance in the SDN environment.

Finally, to avoid data labeling, K-means has been proposed by Kuranage et al. [87]. In this paper, several supervised learning methods were trained and evaluated individually, including SVM, DT, RF, and KNN. The results demonstrate that SVM has the highest accuracy with 96.37%. For this paper, the "IP Network Traffic Flows, Labeled with 75 Apps" dataset was used [88]. However, the features used for the models' training are selected manually;

- **Fine-grained traffic classification**

Fine-grained classification aims to classify the network traffic by application (the exact application generating the traffic) and facilitates the operators to understand users' profiles and hence provide better QoS.

Qazi et al. [89] presented a mobile application detection framework called Atlas. This framework enables fine-grained application classification. The *netstat* logs from the employee devices are collected by the agents to collect ground truth data and are sent to the control plane, which runs the decision tree algorithm (C5.0) to identify the application. Their framework is capable of detecting a mobile application with 94% accuracy for the top 40 Android applications. However, the authors did not present the features used in this framework.

Li and Li [90] presented MultiClassifier, an application classifier, by combining both DPI and ML to do classification in SDN. They try to exploit the advantage of both methods to achieve a high speed with acceptable accuracy. When a new flow arrives, MultiClassifier gives ML a higher priority to classify because it is much faster than DPI. If the reliability of the ML result is larger than a threshold value, its result will be the MultiClassifier results. If not and if DPI does not return "UNKNOWN", MultiClassifier will take the DPI result. This combination succeeded in reaching more than 85% accuracy. Based on the experimental results presented in their paper, we can reveal some conclusions: (i) ML runs four times faster than DPI, and (ii) DPI gives high accuracy. However, the authors did not present the features nor the ML algorithm used for the classification. Moreover, the simulations were not scaled, since they used only two hosts, and thus we cannot confirm the quality of this combination.

Raikar et al. [91] proposed a traffic classification using supervised learning models in the SDN environment. Specifically, a brief comparative analysis of SVM, Naïve Bayes, and KNN has been done, where the accuracy of traffic classification is greater than 90% in all the three supervised learning models. However, the authors used only three applications (SMTP, VLC, HTTP), which is therefore not enough to verify the performance of the models.

In addition, Amaral et al. [62] introduced a traffic classification architecture based on an SDN environment deployed in an enterprise network using ML. In this architecture, the controller collected the flow statistics from the switches, and then the preprocessing step was used, followed by several classifiers individually, including RF, stochastic gradient boosting, and extreme gradient boosting. The accuracy of each application is used as an evaluation metric. However, the study lacks the exploration of the other performance metrics (e.g., f-measure). Additionally, there is no information on the distribution of applications in the dataset. As each application can have several types of flows, like voice and chat, a fine-grained traffic classiication is needed. To solve this issue, Uddin and Nadeem [92] introduced a traffic classification framework called TrafficVision that identifies the applications and their corresponding flow-type in real-time. TrafficVision is deployed on the controller, and one of the kernel modules of TrafficVision is named TV engine, which has three major tasks: (i) collecting, storing, and extracting flow statistics and ground-truth training data from end devices; (ii) building the classifiers from the training data; and (iii) applying these classifiers to identify the application and flow-types in real-time and providing this information to the upper layer application. As a proof of concept, the authors developed two prototypes of "network management" services using the TrafficVision framework. The classification task of the TV engine has two modules, which are application detection using decision tree (C5.0) and flow-type detection using KNN with $K = 3$. Both of these classifiers show more than 90% accuracy. The shortcoming in this paper is the classification of the 40 popular applications and the ignorance of other applications; this can cause problems in an enterprise or university network.

Amaral et al. [93] proposed an application-aware SDN architecture, and the experimental results demonstrate that the semi-supervised classifier outperforms the supervised classifier (random forest classifier) with the same amount of labeled data. This is attributed to the fact that the incorporation of unlabeled data in the semi-supervised training process boosts the performance of the classifiers.

Nakao and Du [94] used deep NN to identify mobile applications. Mobile network traffic was captured from the mobile virtual network operator (MVNO). Five flow features (i.e., a destination address, destination port, protocol type, TTL, and packet size) were selected to train an 8-layer deep NN model. DL achieves 93.5% accuracy for the identification of 200 mobile applications. The features used for classification are selected filter methods in order to train and improve the performance of the DL model.

Wang et al. [95] developed an encrypted data classification framework called DataNet which is embedded in the SDN home gateway. This classification was achieved through the use of several DL techniques, including multilayer perceptron (MLP), stacked autoencoder (SAE), and convolutional neural networks (CNN). They used the "ISCX VPN-nonVPN"

encrypted traffic dataset [96], which consists of 15 encrypted applications (i.e., Facebook, Skype, Hangout, etc.) and more than 200,000 data packets. However, they used only 73,392 flows after adopting the under-sampling method to balance the classes' distribution. This framework helps them to classify the traffic without compromising the security/privacy of services providers or users.

Chang et al. [97] proposed an application for offline and online traffic classification. More specifically, the authors used three DL-based models, including CNN, MLP, and SAE models. Using the Tcpreplay tool, the traffic observations are re-produced and analyzed in an SDN testbed to emulate the online traffic service. The experimental results show that the offline training results have achieved more than 93% accuracy, whereas the online testing prediction achieved 87% accuracy for application-based detection. This may be attributed to the limitation of the processing speed leading to the dropping of statistics packets.

- **Other Approaches for Traffic Classification**

  Video surveillance traffic increased sevenfold between 2016 and 2021 [98]. Due to these changes, multimedia traffic must be managed as efficiently as possible. SDN and ML can be promising solutions to manage video surveillance traffic. Therefore, by applying SDN and ML, we are able to increase efficiency and reduce the cost of management. In this context, Rego et al. [99] proposed an intelligent system for guaranteeing QoS of video traffic in surveillance IoT environments connected through SDN using the AI module, which is integrated into SDN. The traffic classification that detects whether the incoming flow is critical or not is based on the packets sent by the SDN controller to the AI system. This detection is performed through the use of SVM as a classifier and is able to detect critical traffic with 77% accuracy, which is better than other tested methods (i.e., NN or KNN).

  Xiao et al. [60] proposed a real-time elephant flow detection system which includes two main stages. In the first one, suspicious elephant flows are distinguished from mice flows using head packet measurement. In the second stage, the correlation-based filter (CFS) is used to select the optimal features to build a robust classifier. Then, elephant flows are used for improving the classification accuracy, and the decision tree (C4.5) classifies them as real elephant flows or suspicious flows. To maximize the detection rates and minimize the misclassification costs of elephant flows, they used the cost-sensitive decision trees as classifiers.

  Indira et al. [100] proposed a traffic classification method using a deep neural network (DNN) which classifies the packets based on the action (accept/reject). As a proof concept, the performance of DNN is compared with two classifiers, which are SVM and KNN.

  In addition, Abidi et al. [101] proposed a network slicing classification framework. Given network features, the proposed framework tries to find the network slices, such as enhanced mobile broadband (eMBB), massive machine-type communication (mMTC), or ultra-reliable low-latency communication (URLLC) by combined deep belief network and neural network models as well as meta-heuristic algorithms in order to enhance the classification performance.

3.1.2. Public Datasets

The evaluation and the success of ML/DL models depend on the dataset used. In this section, we present some well-known datasets used for network traffic classification.

- **VPN-nonVPN dataset** [96] is a popular encrypted traffic classification dataset. It contains only time-related features and regular traffic, as well as traffic captured over a virtual private network (VPN). Specifically, it consists of four scenarios. Scenario A is used for binary classification in order to indicate whether the traffic flow is VPN or not. Both scenario B and scenario C are classification tasks. Scenario B contains only seven non-VPN traffic services like audio, browsing, etc. Scenario C is similar to Scenario B, but it contains seven traffic services of the VPN version. Scenario D contains all fourteen classes of scenario B and scenario C to perform the 14-classification task;

- **Tor-nonTor dataset**[102] contains only time-related features. This dataset has eight different labels, corresponding to the eight different types of traffic captured, which are browsing, audio streaming, chat, video streaming, mail, VoIP, P2P, and file transfers;
- **QUIC** [103] is released by the University of California at Davis. It contains five Google services: Google Drive, Youtube, Google Docs, Google Search, and Google Music.
- **Dataset-Unicauca** [88] is released by the Universidad Del Cauca, Popayán, Colombia. It was collected through packet captures at different hours during the morning and afternoon over 6 days in 2017. This dataset consists of 87 features, 3,577,296 observations, and 78 classes (Twitter, Google, Amazon, Dropbox, etc.)

### 3.1.3. Discussion

This subsection presents several ML-based solutions for traffic classification in SDN, as summarised in Table 8. It has been divided into two tasks: coarse-grained and fine-grained. Nowadays, as new applications emerge every day, it is not possible to have all the flows labeled in a real-time manner. Therefore, one of the most promising approaches is semi-supervised learning for fine-grained classification, which is closer to reality, as it profits through the use of labeled and unlabeled data. Moreover, DT-based models are the most frequently used for traffic classification where RF shows a good trade-off between performance and complexity. Furthermore, it requires less hyperparameter tuning [104]. Additionally, DL models, as with different domains, verify their effectiveness for network traffic classification. In addition, most of the researchers used a single classical approach for traffic classification. However, using ensemble learning by combining several classifiers achieves better accuracy than any single classifier. Nevertheless, multi-classifier approaches (i.e., ensemble learning) can increase the computational complexity of the model and the classification time. To solve this issue, we can reduce the number of captured packets per flow in order to reduce the classification time. For example, we can use a set of three packets instead of five packets as used in some works [62]. Additionally, based on the literature, each classifier is tested on its environment (i.e., data), and no paper explores all the classifiers with a deep comparison among them.

### 3.2. Traffic Prediction

The objective of network traffic prediction is to forecast the amount of traffic expected based on historical data in order to avoid future congestion and maintain high network quality [105]. It helps to keep the over-provisioning of the resources as low as possible, avoid congestion and decrease communication latency [106,107]. Network traffic prediction can be formulated as the prediction of the future traffic volume ($\hat{y}_{t+l}$) based on the historical and current traffic volumes ($X_{t-J+1}, X_{t-J+2}, \ldots, X_t$). Therefore, the objective of the ML/DL models is to find the parameters that minimize the error between the predicted and observed traffic with respect to $X_{t-J+1}, X_{t-J+2}, \ldots, X_t$ (Equations (1) and (2)).

$$W^* = \underset{W^*}{\operatorname{argmin}}\, L(y_{t+l}, \hat{y}_{t+l}; W^*) \tag{1}$$

$$\hat{y}_{t+l} = f([X_{t-J+1}, X_{t-J+2}, \ldots, X_t]) \tag{2}$$

where $y_{t+l}$ and $\hat{y}_{t+l}$ are the observed and predicted value at time $t + l$, respectively, $f(.)$ is the activation function, $L$ is the loss function, and $W^*$ is the optimal set of parameters.

### 3.2.1. Existing Solutions

In this section, we present the recent works concerning traffic prediction. Kumari et al. [108] presented a framework for SDN traffic prediction. To do so, two prediction models, namely ARIMA and SVR (support vector regression) have been used. The results showed that SVR outperforms the ARIMA method, where the average performance improvement is 48% and 26%. However, the temporal and spatial variation of network traffic made the accurate prediction of flows challenging. Consequently, learning highly complicated patterns requires more complicated models, like deep learning-based models.

**Table 8.** Summary of reviewed papers on ML application for traffic classification in SDN.

| Classification Level | Ref. | ML/DL Algorithm | Dimensionality Reduction | Dataset Output | Controller |
|---|---|---|---|---|---|
| Coarse-grained classification | [61] | DPI and Laplacian SVM | Wrapper method | Voice/video conference, interactive data, streaming, bulk data transfer | N/A |
| | [78] | Feedforward, MLP | - | Instant message, stream, P2P, HTTP, FTP | Floodlight |
| | [79] | Spectral clustering | N/A | HTTP, SMTP, SSH, P2P, DNS, SSL2 | Floodlight |
| | [59] | SVM | PCA | DDoS attacks, FTP, video streaming | Floodlight |
| | [63] | Stacked Autoencoder | AE | Bulk, database, interactive, mail, services, WWW, P2P, attack, games, multimedia | N/A |
| | [86] | Heteroid tri-training (SVM, KNN, and Bayes classifier) | N/A | Voice, video, bulk data, interactive data | N/A |
| | [87] | SVM, decision tree, random forest, KNN | - | N/A | RYU |
| | [80] | C4.5, KNN, NB, SVM | Filter and wrapper method | WWW, mail, bulk, services, P2P, database, multimedia, attack | N/A |
| | [83] | CNN | - | WWW, mail, FTP-control, FTP-data, P2P, database, multimedia, services, interactive, games | N/A |
| | [81] | DT, RF, GBM, LightGBM | N/A | Web browsing, email, chat, streaming, file transfer, VoIP, P2P | RYU |
| | [82] | Ensemble classifier (KNN, SVM, RF, AdaBoost, Boosting, XGBoost) | - | Video, voice, bulk data transfer, music, interactive | N/A |
| Fine-grained classification | [85] | Deep learning | - | WWW, mail, FTP-control, FTP-pasv, FTP-data, attack, P2P, database, multimedia, services | POX |
| | [90] | DPI and classification algorithm | N/A | N/A | Floodlight |
| | [89] | Decision tree (C5.0) | N/A | Top 40 Android applications | N/A |
| | [62] | Random forest, stochastic gradient boosting, XGBoost | PCA | Bittorent, Dropbox, Facebook, HTTP, Linkedin, Skype, Vimeo, Youtube | HP VAN |
| | [92] | Decision tree (C5.0), KNN | - | The top 40 most popular mobile applications | Floodlight |
| | [95] | MLP, SAE, CNN | AE | AIM, email client, Facebook, Gmail, Hangout, ICQ, Netflix, SCP, SFTP, Skype, Spotify, Twitter, Vimeo, Voipbuster, Youtube | N/A |
| | [94] | Deep Learning | Filter method | 200 mobile applications | N/A |
| | [93] | Random forest | - | Bittorent, Dropbox, Facebook, HTTP, Linkedin, Skype, Vimeo, Youtube | HPE VAN |
| | [97] | MLP, CNN, SAE | - | Facebook, Gmail, Hangouts, Netflix, Skype, Youtube | Ryu |
| | [91] | SVM, KNN, Naive Bayes | N/A | SMTP, HTTP, VLC | POX |
| Others | [99] | SVM | | Video surveillance traffic in IoT environment (critical or non-critical traffic) | N/A |
| | [60] | Decision tree (C4.5) | Filter method | Elephant flow, mice flow | Floodlight |
| | [100] | DNN | - | Action (accept/reject) | N/A |
| | [101] | Deep belief network, neural network | - | Enhanced mobile broadband slice, massive machine-type communications slice, ultra-reliable low-latency communication slice | N/A |

From the recent literature, we can find representative DL methods that are frequently used for network prediction tasks in the SDN environment. For example, Azzouni and Pujolle [109] proposed a framework called neuTM to learn the traffic characteristics from historical traffic data and predict the future traffic matrix using LSTM (long short-term memory). They implemented their framework and deployed it on SDN, then trained it on a real-world dataset using different model configurations. The experimental results

show that their model converges quickly and can outperform linear forecasting models and feed-forward deep neural networks. Alvizu et al. [110] applied a neural network model to predict network traffic load in a mobile network operator. The prediction results are used to make online routing decisions. The authors in this work train a real dataset for mobile network measured during November and December 2013 in Milano, Italy. Similarly, based on ANN, Chen-Xiao and Ya-Bin [111] proposed a solution to keep network load balanced using SDN. This solution aims to select the least-loaded path for newcomer data flow. They take advantage of the global view of SDN architecture to collect four features of each path, which are bandwidth utilization ratio, packet loss rate, transmission latency, and transmission hop, and use ANN to calculate the load condition of each path.

In addition, network traffic prediction is even important for the IoT network given the huge amount of IoT devices widely used in our daily life. To improve the transmission quality for such a network, an efficient deep learning-based traffic load prediction algorithm to forecast future traffic load was proposed by Tang et al. [112]. The authors presented the performance of their algorithms with three different systems (centralized SDN system, semi-centralized, and a distributed conventional control system without centralized SDN). For the centralized SDN system, the authors used $M$ deep CNN where each deep CNN is only used to predict the traffic load of one switch. In each switch, the traffic load consists of two parts: (i) the relayed traffic flow from other switches, and (ii) the integrated traffic flow composed by the sensing data from devices. However, for the semi-centralized SDN system, each switch uses just a deep belief network to predict the traffic generated by connected sensing devices; then, centralized SDN uses Deep CNN to make the final prediction. Based on the obtained results, the accuracy with a centralized SDN system is always better than the two other systems (more than 90%).

Due to the volatile nature of network traffic in smaller time scales, Lazaris et al. [113] focused on network traffic prediction over short time scales using several variations of the LSTMs model. To evaluate the performance of their solution, the authors used real-life traffic (CAIDA Anonymized Internet Traces 2016 Dataset) [114]. The experimental results demonstrate that the LSTM models perform much better than the ARIMA models in all scenarios (e.g., various short time scales).

Le et al. [115] proposed a DL-based prediction application for an SDN network. Specifically, the authors have used three RNN variants models, which are long short-term memory (LSTM), gated recurrent units (GRU), and BiLSTM. Their experiments show that all three models, especially the GRU model, perform well but still produce significant predicting errors when the traffic is suddenly shot. However, BiLSTM consumes the most resources in training and predicting.

Additionally, to ensure dynamic optimization of the allocation of network resources in a proactive way, Alvizu et al. [106] proposed machine-learning-based traffic prediction. Specifically, the GRU model was used in order to predict the mobile network traffic matrix in the next hour.

Moreover, with the scale and complexity expected for the networks, it is essential to predict the required resources in a proactive way. To do so, network traffic forecasting is the need of the hour. In this context, Ferreira et al. [116] used both linear and non-linear forecasting methods, including machine learning, deep learning, and neural networks, to improve management in 5G networks. Through these forecasting models, a multi-slice resource management approach has been proposed, and the experimental results show that it is possible to forecast the slices' needs and congestion probability efficiently and, accordingly, maintain the QoS of the slice and the entire network.

Last but not least, since the ML/DL-based models need continuous training because the network situation changes quickly, sending all raw data to a central entity can slow the convergence of the models as well as introduce network congestion. To solve these issues, Sacco et al. [75] used the LSTM model in a collaborative way in order to predict the future load to optimize routing decisions (i.e., select the best path). More specifically, the authors used federated architecture with a multi-agent control plane, where each controller trains

the LSTM model locally then sends only its model parameters to the Cloud for global aggregation (Figure 7).

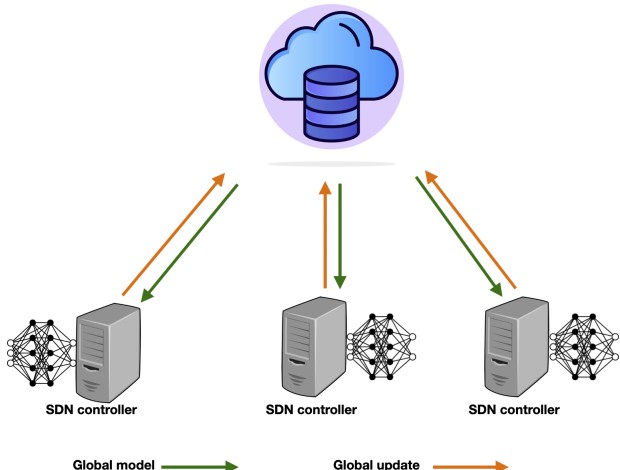

**Figure 7.** A federated learning approach with SDN-enabled edge networks [75].

3.2.2. Public Datasets

The dataset is an important component in the process of traffic prediction, as it has a direct impact on the accuracy and applicability of the model. Here, we identify some public datasets that can be used for network traffic prediction.

- **Abilene dataset** [117] contains the real trace data from the backbone network located in North America consisting of 12 nodes and 30 unidirectional links. The volume of traffic is aggregated over slots of 5 min starting from 1 March 2004 to 10 September 2004;
- **GEANT** [118] has 23 nodes and 36 links. A traffic matrix (TM) is summarized every 15 min starting from 8 January 2005 for 16 weeks (10,772 TMs in total);
- **Telecom Italia** [119] is part of the "Big Data challenge". The traffic was collected from 1 November 2013 to 1 January 2014 using 10 min as a temporal interval over Milan. The area of Milan is divided into a grid of $100 \times 100$ squares, and the size of each square is about $235 \times 235$ m. This dataset contains three types of cellular traffic: SMS, call, and Internet traffic. Additionally, it has 300 million records, which comes to about 19 GB.

3.2.3. Discussion

In the above subsection, several ML-based solutions for traffic prediction in SDN have been presented, as summarized in Table 9. Mobile communication faces many challenges when the number of smart devices increases, and hence, the load in the network can lead to network congestion. In this context, ML models can be used to predict the network situation and help the operator to anticipate exceptional resource demand or reduce resources when no longer needed. Traffic prediction is considered a regression problem (i.e., dataset outputs have continuous values). Therefore, supervised learning can be used for traffic prediction. Based on the literature, DL algorithms, especially LSTM, are widely adopted and can be considered a promising solution for network traffic prediction. However, building an LSTM model does not mean successful results, since several factors can affect the performance of the model, such as the model hyperparameters and the size of the dataset used during the training process. Additionally, the model needs continuous training, since the network situation changes quickly. Moreover, the exchange of information between the controller and the forwarding devices can overload the system and can pose security and data privacy problems. Therefore, federated learning seems to be an efficient solution. This method can optimize the communication between SDN and the involved forwarding devices by keeping the data where it was generated. This leads to preserving the bandwidth for the application traffic, reducing costs of data communication, and ensuring data security. Thus, more attention should be given to this research direction.

**Table 9.** Summary of reviewed papers on ML applications for network traffic prediction in SDN.

| Ref. | ML/DL Model | Controller | Contribution |
|---|---|---|---|
| [109] | LSTM | POX | The authors proposed a traffic prediction framework and deployed it on SDN using a real-world dataset under different model configurations. |
| [110] | ANN | N/A | The authors used the prediction results to make online routing decisions. |
| [111] | ANN | Floodlight | The proposed system tries to benefit from the global view of SDN controller in order to collect the bandwidth utilization ratio, packet loss rate, transmission latency, and transmission hop of each path. Then, using the ANN model, it predicts the load condition of each path. |
| [112] | Deep CNN | N/A | The authors presented the performance of their systems with three different systems, which are a centralized SDN system, a semi-centralized SDN system, and a distributed conventional control system without centralized SDN. |
| [106] | GRU-RNN | N/A | The authors used (GRU-RNN) in order to predict the mobile network traffic matrix in the next hour. |
| [108] | SVR | N/A | The authors demonstrate that SVR outperforms the ARIMA method; the average performance improvement is 48% and 26%. |
| [113] | LSTM | N/A | The authors focused on network traffic prediction for short time scales using several variations of the LSTM model. |
| [75] | LSTM+FL | Floodlight | The authors used an LSTM model and federated learning in order to predict the future load to optimize routing decisions and at the same ensure data privacy as well decrease the exchange message between the SDN controller. |
| [115] | LSTM, BiLSTM, GRU | POX | The authors proposed a comparative analysis between three RNN-based models: LSTM, GRU, and BiLSTM. |
| [116] | Shallow ML models, DL models, ensemble learning | N/A | The authors take advantage of the forecasting results in order to manage the network slice resource. |

### 3.3. Network Security

Despite the advantages of SDN, its security remains a challenge. More specifically, the centralized control plane of SDN can be a point of vulnerability [120]. An intrusion detection system (IDS) is one of the most important network security tools due to its potential in detecting novel attacks [121]. According to Base and Mell [122], intrusions are defined as *"attempts to compromise the confidentiality, integrity, or availability of a computer or network, or to bypass the security mechanisms of a computer or network"*.

#### 3.3.1. Existing Solutions

Recently, ML/DL based models have been widely used for intrusion and attack detection. In this section, we present a literature review of these models for IDS (Table 10).

ML-based models are widely applied for network IDS. For example, Latah et al. [123] proposed a comparative analysis between different conventional ML-based models for the intrusion detection task using a benchmark dataset (NSL-KDD dataset). The features used by these classifiers were extracted and reduced after the application of the PCA method. The results demonstrate that using PCA enhances the detection rate of the classifiers, as well as the DT approach, showing the best performance in terms of all the evaluation metrics.

Additionally, as SDN and ML-based models are considered the enablers of the realization of the 5G network, Li et al. [124] proposed an intelligent attack classification system in an SD-5G network using RF for feature selection and AdaBoost for attack classification with the selected features. The 10% of the KDD Cup 1999 dataset was used to evaluate the proposed system. The results demonstrate that, using the selected features, the classifier can better differentiate the attack traffic and produce a low overhead system. Similarly, Li et al. [125] proposed a two-stage intrusion detection and attack classification system in an SD-IoT network. In the first stage, they selected the relevant features, which were used in the second stage for attack anomaly detection. The experiments on the KDD Cup 1999 dataset show that random forest achieved a higher accuracy with an acceptable complexity time compared to the existing approaches.

However, as conventional ML models have difficulties detecting attacks in large-scale network environments [126], DL models have been used. They can give better

performance for intrusion detection, as with network traffic classification and prediction (Sections 3.1 and 3.2) without the need to extract the features manually. In this context, Tang et al. [45] proposed a DNN model for anomaly detection using only six features that can be easily obtained in an SDN environment. The experimental results on the NSL-KDD dataset show that the deep learning approach outperforms the conventional ML models. Then, Tang et al. [127,128] proposed, for the first time, the GRU-RNN model for anomaly detection in SDN, which is an extension of their previous work [45]. This model has the ability to learn the relationship between current and previous events. Using the NSL-KDD dataset, the GRU-RNN model achieved 89% accuracy using the 6 features. Additionally, the experimental results demonstrate that the GRU-RNN model outperforms VanilaRNN, SVM, and DNN.

To also improve the detection rate of the attack or the intrusion, some researchers have started to combine DL-based models and conventional or classical ML models. More specifically, they take advantage of the benefits of the deep features extracted through deep learning, which can be used with conventional models for classification or attack detection. In this context, Elsayed et al. [129] combined CNN architecture with conventional ML-based models, including SVM, KNN, and RF. Specifically, CNN extracts the deeper representations from the initial features, while the classification task is performed through SVM, KNN, and RF. To evaluate the performance of their approach, the authors used a novel dataset, called *InSDN*, which is specific to an SDN network [130], along with UNSW-NB15, and CIC-IDS2018 datasets. The results demonstrate the potential of CNN for anomaly detection even with a few amount of features (9 features). Also, the combination of CNN and SVM, KNN, and especially the RF algorithm provide higher performance compared to a single CNN.

Unlike the centralized architecture, distributed SDN can deploy multiple IDS for the active detection of attacks. In contrast to the other IDS approaches, Shu et al. [131] proposed a collaborative IDS based on distributed SDN in VANETs, called *CIDS*. In other words, CIDS enables all the distributed SDN controllers to collaboratively train a stronger detection model based on the whole network flow. To do so, generative adversarial networks (GAN) have been used, wherein a single discriminator is trained on the Cloud server and several generators are trained on the SDN controllers. Using the KDD99 and NSL-KDD datasets, the evaluation results show that the proposed method achieves better performance as compared to centralized detection methods.

Although distributed SDNs are trained on richer data, they are limited in terms of collaboration. Consequently, the FL started to attract researchers since there is a need for collaboration among the domain. In this context, Thapa et al. [74] proposed an FL model for the detection and mitigation of ransomware attacks in the healthcare system called *FedDICE*. Specifically, FedDICE integrates FL in an SDN environment to enable collaborative learning without data exchange. The performance of the FedDICE framework is similar to the performance achieved by centralized learning. Similarly, Qin et al. [132] used FL for intrusion detection in the SDN environment. They combined FL and binarized neural networks for intrusion detection using programmable network switches (e.g., the P4 language). Using programmable switches and FL enables the incoming packet to be classified directly in the gateway instead of forwarded to the edge controller or Cloud server. The results demonstrate that FL leads to more accurate intrusion detection compared with training each gateway independently.

- **DDoS attack detection/classification**

A DDoS attack is a complex form of denial of service attack (DoS). It generates a huge amount of source IP address and destination IP address packets, and in turn, the switches send a large number of packets to the controller. This may exhaust the networking, storage, and computing resources of the controller [133]. Therefore, detection and mitigation of DDoS attacks in real-time is necessary. As a result, several ML models have been used for DDoS attack detection in the SDN environment. In this context, Ahmad et al. [134] evaluated different well-known models, including SVM, naive Bayes, DT, and logistic

regression. The results show that the SVM performed better than other algorithms, since the classifiers were used for the binary classification scenarios.

Chen et al. [135] used XGBoost as a detection method in a SDN-based Cloud network. Then, they evaluated the efficiency of XGBoost through the Knowledge Discovery and Data mining (KDD) Cup 1999 dataset. The evaluation was done considering a binary classification scenario (DDoS/benign traffic). A final comparison was made between XGBoost, Random Forest, and SVM. Additionally, to detect the anomaly in the data plane level and to minimize the load on the control plane, Da el al. [107] proposed a framework for the detection, classification, and mitigation of traffic anomalies in the SDN environment called *ATLANTIC*. It consists of two phases, which are (i) the lightweight phase and (ii) the heavyweight phase. The lightweight phase is responsible for anomaly detection by calculating the deviation in the entropy of flow tables, whereas the heavyweight phase uses the SVM model for anomaly classification.

Recently, like the other domains, DL models have been used for DDoS attack detection. For example, Niyaz et al. [136] proposed a DL-based system for DDoS attack detection in an SDN environment. Specifically, SAE has been used for feature reduction in an unsupervised manner. Then, the traffic classification was performed with the softmax layer in the scenario of two classes (DDoS/benign traffic) and the eight classes, which included normal traffic and seven kinds of DDoS attacks. The experimental results show that the SAE model achieved higher performance compared to the neural network model using their own private dataset.

Other DL models have been used for DDoS attack detection like CNN, RNN, and LSTM. For example, Li et al. [137] used CNN, RNN, and LSTM as neural network models to realize the detection of DDoS attacks in the SDN environment. The evaluation and training tasks of the used models were done on the ISCX2012 dataset. Similarly, Haider et al. [138] used the same DL-based model for DDoS attack detection using a benchmark dataset (CICIDS2017 dataset). More specifically, four DL-based models were applied in an ensemble (RNN+RNN, LSTM+LSTM, CNN+CNN) or in a hybrid way (e.g., RNN+LSTM). The results show the capability of DL models and especially the ensemble CNN model in detecting DDoS very well. Ensemble CNN provides high detection accuracy (99.45%); however, it takes more time for the training and classification tasks.

Moreover, Novaes et al. [120] proposed a detection and mitigation framework against DDoS attacks in SDN environments. To do so, they used the generative adversarial Network (GAN) model in order to make the proposed framework less sensitive to adversarial attacks. The experiments were conducted on emulated data and the public dataset CICDDoS 2019, where the GAN obtained superior performance compared to the CNN, MLP, and LSTM models.

Although the DL-based models are efficient for the DDoS attack detection/classification, combining deep and conventional ML models can take advantage of both techniques and hence further improve the performance of the security system. Krishnan et al. [139] proposed a hybrid ML approach by combining DL and conventional ML models called *VARMAN*. AE was used to generate 50 new reduced features from the CICIDS2017 dataset as well as to optimize the computation and memory usage. Then, random forest acts as the main DDoS classifier. The experiments demonstrate that VARMAN outperforms the deep belief networks model, and this is attributed to the combination of different models.

### 3.3.2. Public Datasets

As the performance of the IDS techniques relies on the quality of the training datasets, in this subsection, we review the publicly widely used datasets to evaluate the performance of the ML/DL-based model for intrusion detection.

- **KDD'99** [140] is one of the most well-known datasets for validating IDS. It consists of 41 traffic features and 4 attack categories besides benign traffic. The attack traffic is categorized into denial of service (DoS), remote to local (R2L), user to root (U2R), or probe attacks. This dataset contains redundant observations, and therefore, the trained model can be biased towards the more frequent observations;

- **The NSL-KDD dataset** [141] is the updated version of the KDD'99 dataset. It solves the issues of the duplicate observations. It contains two subsets, a training set and testing set, where the distribution of attack in the testing set is higher than the training set;
- **The ISCX dataset** [121] was released by the Canadian Institute for Cybersecurity, University of New Brunswick. It consists of benign traffic and four types of attack, which are brute force attack, DDoS, HttpDoS, and infiltrating attack;
- **Gas pipeline and water storage tank** [142] is a dataset released by a lab at Mississippi State University in 2014. They proposed two datasets: the first is with a gas pipeline, and the second is for a water storage tank. These datasets consist of 26 features and 1 label. The label contains eight possible values, benign and seven different types of attacks, which are naive malicious response injection, complex malicious response injection, malicious state command injection, malicious parameter command injection, malicious function command injection, reconnaissance, and DoS attacks;
- **The UNSW-NB15 dataset** [143] is one of the recent datasets; it includes a simulated period of data which was 16 h on 22 January 2015 and 15 h on 17 February 2015. This dataset has nine types of attacks: fuzzers, analysis, backdoors, DoS, exploits, generic, reconnaissance, shellcode, and worms. It contains two subsets, a training set with 175,341 observations and a testing set with 82,332 observations and 49 features;
- **The CICDS2017 dataset** [144] is one of the recent intrusion detection datasets released by the Canadian Institute for Cybersecurity, University of New Brunswick. It contains 80 features and 7 types of attack network flows: brute force attack, heartbleed attack, botnet, DoS Attack, DDoS Attack, web attack, infiltration attack;
- **The DS2OS dataset** [145] was generated through the distributed smart space orchestration system (DS2OS) in a virtual IoT environment. This dataset contains benign traffic and seven types of attacks, including DoS, data type probing, malicious control, malicious operation, scan, spying, and wrong setup. It consists of 357,952 observations and 13 features;
- **The ToN-IoT dataset** [146] was put forward by the IoT Lab of the UNSW Canberra Cyber, the School of Engineering and Information Technology (SEIT), and UNSW Canberra at the Australian Defence Force Academy (ADFA). It contains traffic collected from the Internet of Things (IoT) and industrial IoT devices. It contains benign traffic and 9 different types of attacks (backdoor, DDoS, DoS, injection, MITM, password, ransomware, scanning, and XSS) with 49 features;
- **The IoT Botnet dataset** [147] uses the MQTT protocol as a communication protocol. It contains 83 features and 4 different types of attacks, including DDoS, DoS, reconnaissance, and theft, along with benign traffic;
- **The InSDN dataset** [130] is a recent dataset and the first one generated directly from SDN networks. It consists of 80 features and 361,317 observations for both normal and attack traffic. It covers different types of attack types, including DoS, DDoS, probe, botnet, exploitation, password guessing, web attacks, and benign traffic (HTTPS, HTTP, DNS, Email, FTP, SSH) that can occur in the SDN environment.

### 3.3.3. Discussion

Different ML/DL-based models have been used for intrusion detection/classification. Despite the performance of DL models, many solutions use conventional ML models (shown from Table 10) due to their simplicity. However, these models are usually accompanied by some dimensionality reduction methods. Additionally, as seen in the literature review, the detection of DDoS attacks has attracted many researchers. In other words, since the SDN controller represents the central brain of the network that stores and processes the data from all forwarding devices, it can be targeted by DDoS attacks. Moreover, FL can be used to improve the security and privacy of the end-users by keeping the traffic at the level of the data plane without damaging the controller. It is able to involve a massive number of forwarding devices which are important for training DL models. Therefore, by only sharing the local update

of the global model between the forwarding devices and the controller, the communication overhead may be reduced, hence speeding up the attack detection/classification.

**Table 10.** Summary of reviewed papers on ML applications for network security in SDN.

| Ref. | Category | ML/DL Model | Dimensionality Reduction | Controller | Contribution |
|---|---|---|---|---|---|
| [45] | Intrusion detection | DNN | - | N/A | The first application of the DL approach for intrusion detection in the SDN environment. |
| [127] [128] | Intrusion detection | GRU-RNN | - | POX | The authors achieve a good performance using the GRU-RNN model with only six features. |
| [135] | DDoS detection | XGBoost | Filter method | POX | The XGBoost algorithm has strong scalability and higher accuracy and a lower false positive rate than random forest and SVM. |
| [136] | DDoS classification | SAE | AE | POX | Before the DDoS attack detection task, the authors used SAE for feature reduction in an unsupervised manner. |
| [120] | DDoS classification | GAN | - | Floodlight | The authors used GAN to make their system more efficient against adversarial attacks. |
| [125] | Attack classification | RF | Filter method | N/A | The authors used an upgrade metaheuristic algorithm for feature selection and RF as a classifier. |
| [124] | Attack classification | AdaBoost | Wrapper method | N/A | The authors demonstrate that using the feature selection method improves the attack classification task as well as produces a low overhead system. |
| [137] | DDoS detection | LSTM, CNN, RNN | - | N/A | The authors have compared different DL-based models. |
| [138] | DDoS detection | CNN | - | N/A | The authors proposed an ensemble CNN model for the DDoS attack detection and compared its performance with other known DL-based models using a benchmark dataset. |
| [139] | DDoS classification | AE+RF | AE | Ryu | The authors proposed a hybrid ML approach by combining deep learning and conventional ML models. The AE is used to extract a reduced version of the initial features, and the RF acts as the main classifier of the system. |
| [129] | Attack classification | CNN+SVM, CNN+RF, CNN+DT | CNN | N/A | The authors demonstrate the potential of CNN for anomaly detection even with a few amount of features (9 features). Additionally, the combination of CNN and SVM, KNN, and especially the RF algorithm enhances the detection rate and provides a higher performance compared to the single CNN. |
| [134] | DDoS detection | SVM, Naive-Bayes, DT, and Logistic Regression | N/A | POX | The authors compare the performance of different well-known ML-based models, including SVM, Naive-Bayes, DT, and logistic regression. |
| [123] | Intrusion detection | DT, RF, AdaBoost, KNN, SVM | PCA | N/A | The authors focused on a comparative analysis between different conventional ML-based models for the intrusion detection task using a benchmark dataset (NSL-KDD dataset). |
| [107] | DDoS classification | SVM | PCA | Floodlight | The authors proposed a framework for detection and classification called *ATLANTIC*. The attack detection is performed by calculating the deviations in the entropy of flow tables, whereas the attack classification was done by the SVM model. |
| [131] | Attack classification | GAN | - | N/A | The authors proposed an IDS based on distributed SDN in VANETs in order to enable all the distributed SDN controllers to collaboratively train a stronger detection model based on the whole network flow. |
| [74] | Ransomware attacks | Logistic Regression+FL, FNN+FL | - | N/A | The authors proposed a federated learning model in an SDN environment to enable collaborative learning for detection and mitigation of ransomware attacks in healthcare without data exchange. |
| [132] | Intrusion detection | BNN+FL | - | N/A | The authors combined FL and binarized neural networks for intrusion detection using programmable network switches (e.g., P4 language). |

## 4. Research Challenges and Future Directions

Research works on the use of ML techniques with networks have produced a multitude of novel approaches. The application of ML/DL models to networking brings several use cases and solutions. However, these solutions suffer from some issues and challenges in practice [50,148]. Many important factors should be considered when designing an ML-based system in an SDN environment. In this context, this section discusses the key challenges and the issues of using ML/DL algorithms for network management in the SDN environment. Then, we present some research opportunities in related areas.

**Imbalanced data samples**: Class imbalances are one of the critical problems in traffic classification/intrusion detection. For example, most of the time, the samples of the anomaly are rare, and there is a great imbalance between the amount of several applications' traffic (i.e., Google, Facebook, Skype) compared to other traffic. A two-class dataset is said to be imbalanced if one of the classes (the minority one) is represented by a very small number of instances in comparison to the other (majority) class. The classifier tends to be biased toward the larger groups that have more observations in the training sample [149]. Therefore, ignoring the unequal misclassification risk for different groups may have a significant impact on the practical use of the classification [23]. Several methods have been proposed as solutions to imbalance class problems, such as under-sampling, over-sampling, synthetic minority over-sampling techniques (SMOTE), or using boosting methods [150].

**Identifying DL architecture**: Engineering the right set of features for a given scenario is often key to the success of a machine-learning project and is not an easy-to-solve question. Therefore, one of the challenges of ML is to automate more and more of the feature engineering process. DL is currently the most active topic of ML. It can process raw data and generate relevant features autonomously. These features are automatically extracted from the training data by DL. However, it is computationally demanding, especially if computing resources are limited. DL has many hyperparameters and their number grows exponentially with the depth of the model. Therefore, it requires a lot of tuning, and there is no clear mathematical proof to interpret its architecture. Finding suitable architecture (i.e., number of hidden layers) and identifying optimal hyperparameters (i.e., learning rate, loss function, etc.) are difficult tasks and can influence the model performance.

**Available datasets**: Note that the model training performance is highly dependent on the dataset. Therefore, it is one important component for ML/DL models training, as it has a direct impact on the accuracy and applicability of classifiers [86]. Although there are a few recent public datasets available for traffic classification [88,96], there is no commonly agreed-upon dataset for most traffic-related classification problems. Additionally, choosing the right dataset is difficult because ML models can only identify applications based on what it has known (existing in the training set). ML algorithms, and especially DL models, can further benefit from training data augmentation. This is indeed an opportunity for using ML/DL algorithms to improve QoS, as the networks generate tremendous amounts of data and the intensive growth of applications on the Internet. However, due to privacy and lack of shareable data, there is difficulty in progressing on traffic classification. To solve these issues, distributed learning and, especially, FL have started to attract more and more researchers for network traffic management. Moreover, due to the huge number of applications on the Internet, the dataset cannot contain all of them. Therefore, the amount of available data could still be insufficient to train ML algorithms effectively, especially DL algorithms. For these reasons, many researchers try to use old datasets to test their solutions. However, this solution is inefficient, as the behaviors of modern applications change, and their complexity increases every day. Furthermore, most network data are unlabeled; hence, fine-grained classification becomes impractical, and it is unrealistic to achieve low complexity classification based on unsupervised learning.

**Selection of ML models**: ML/DL models are constructed and tested in their environment (i.e., dataset, features). However, average high performance does not guarantee high performance with other problems. Additionally, the choice of the appropriate model is a daunting task, since it depends on many parameters such as the types of data, the size

of the data samples/features, the time, and computing resource constraints, as well as the type of prediction outcomes. Therefore, in light of the diversity of ML/DL models, choosing the appropriate one is a difficult task because a single wrong decision can be extremely costly.

**Selection of flow features**: Besides the selection of the suitable ML model, the performance of this model relies on the collection of features used for the flow classification. Thus, one of the most challenging aspects of flow-based techniques is how to select the appropriate flow of statistical features. Researchers try to only use features which are easy to be obtained with the SDN controller to train the model, but this solution can decrease the performance of ML models [45]. Additionally, some features can have a continuous or discrete value. However, several ML models cannot learn with two types of features at the same time, such as SVM and XGBoost.

**Scalable traffic classification**: In recent years, we have observed enormous growth in Internet traffic. Increased traffic volume makes traffic classification tasks computationally intensive and imposes several challenges to implement in real-time. Distributed architecture can solve this issue and rapidly query, analyze, and transform data at scale. Spark performs 100x faster computation than Hadoop [151]. It also provides a fast machine learning library, MLib, that leverages high-quality learning algorithms. However, based on Table 8, no study has carried out an experimental evaluation on the scalability of ML-based traffic classification in SDN. Moreover, FL can be used in order to take advantage of the end-user and edge devices by training the model locally in a distributed way on such devices.

**Variance and bias**: The most important evaluation criterion for a learner is its ability to generalize. Generalization error can be divided into two components, which are variance and bias. Variance defines the consistency of a learner's ability to predict random things (over-fitting), and bias describes the ability of a learner to learn the wrong thing (under-fitting) [18]. A learner with the lowest bias, however, is not necessarily the optimal solution, because the ability to generalize from training data is also assessed by a second parameter termed variance. A major challenge for ML/DL models is to optimize the trade-off between bias and variance. Therefore, to prevent this problem, we can split our data into three subsets, which are training, validation, and testing. The training set is used for the learning task, validation helps to find the parameters of the learner, and then the testing set is used to evaluate the performance of the constructed model. Additionally, another method named K-fold cross-validation can be used to avoid the problem of over-fitting. In this method, the data is divided into $k$ folds; then, $k - 1$ folds are used as the training set, and the remaining fold is used as the test set. However, it is expensive in terms of training time and computation.

**Network Slicing**: As presented in the literature, the definition of a slice is based on the QoS requirement proposed by NGMN, and no intelligence has been applied in this context. Currently, new online applications are emerging every day, and network behavior is constantly changing. Additionally, the explosion of smart devices and the surge in traffic will need different requirements for their performance. Therefore, the ML concepts can help to find more behaviors and better slices even with a single network infrastructure.

**Knowledge transfer in multiple SDN controllers:** The model trained by some SDN controllers can capture common features with other SDN controllers (e.g., domain). This means the data can be transferable between the domain and hence can provide a more general model. For this purpose, transfer learning can be used in order to transfer the knowledge between the different SDN controllers. It helps the model to avoid being trained from scratch, thus accelerating the model convergence and solving the problem of insufficient training data [152].

**Privacy**: In recent years, privacy has been one of the most important concerns in the network domain. ML models, especially DL models, may benefit from training data augmentation. However, the end-users or clients refuse to provide their data due to the risk of data misuse or inspection by external devices. To solve this issue and guarantee that training data remains on its owners, a distributed learning technique like FL can be a

promising solution. It enables the operators to benefit through the data of clients without sacrificing their privacy. In this context, FL can be the appropriate approach for network traffic management, especially with SDN architecture, as it reduces the data exchange between the data plane and SDN controller. Therefore, it guarantees data confidentiality, minimizes costs of data communication, and relieves the burden on the SDN controller. Additionally, it can be used with network slicing for user prediction for each slice to enable the local data training without the need of sharing the data between the slices.

## 5. Conclusions

In this survey, we examined the integration of machine and deep learning in next generation networks, which is becoming a promising topic and enables intelligent network traffic management. We first provide an overview of ML and DL along with SDN in which the basic background and motivation of these technologies were detailed. More specifically, we have presented that SDN can benefit from ML and DL approaches. Furthermore, we have summarized basic concepts and advanced principles of several ML and DL models, as well as their strengths and weaknesses. We have also presented dimensionality reduction techniques (i.e., feature selection and extraction) and their benefits with the conventional ML models. In addition, we have shown how ML/DL models could help various aspects of network traffic management, including network traffic classification, prediction, and network security. However, the combination of SDN and ML can suffer from various issues and challenges. Therefore, we discussed those which need the researcher's attention both in industry and academia. In summary, research on the integrated SDN and ML for intelligent traffic network management in next-generation networks is promising, and several challenges lay ahead. This survey attempts to explore essential ingredients related to the integration of ML in SDN and aims to become a reference point for the future efforts of researchers who would like to delve into the field of intelligent traffic networks and their applications.

**Funding:** This research received no external funding.

**Data Availability Statement:** Not Applicable. The study does not report any data.

**Conflicts of Interest:** The authors declare no conflict of interest.

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
