# Peer review of "Intelligent Traffic Management in Next-Generation Networks"

_futureinternet, doi:10.3390/fi14020044_

Round 1

Reviewer 1 Report

The review is interesting and devoted to the actual topic of intelligent traffic management. Below, I suggest several comments that can improve the understanding of this review.
1) The authors should check abbreviations throughout the text. For example SDN abbreviation appears suddenly in line 36 without any definition. On the contrary ML definition appears twice in lines 6 and 118. Then the list of acronyms is presented in table 2.
2) It should be clarified why the authors have chosen Federated Learning and Intrusion Detection as the main components of the study in Fig.1.
3) Fig. 2 requires clarifications too. For example, the left side of the picture presents a scheme for classical ML approaches including training steps. The right side of that picture is presented without ANN/CNN training steps, why?
4) Section 3 title contains the term "Softwarized". I think this term is not very common to be included in the section title.
5) The authors should think about the correctness of Fig.6, because feature modeling depends only on test samples and CNN classifier depends on training samples. Is it correct?
Overall, I think the paper is worth accepting after the minor corrections above.

Author Response

We would like to thank you for the valuable comments that we have taken into account in revising our paper.  You can find it in the attachment of the supplement file.

Reviewer 2 Report

 The proposal refers to a relevant practical problem concerning the resolving the Intelligent traffic management in networks.

It is a long survey article involving a very large number of references (155).

The proposal goals are well specified: the use of artificial intelligence in traffic management.

The involved methods (machine learning, deep learning) are well justified.

The first part contains a large review related to general concepts concerning machine learning and deep learning.

The second part includes the survey of the applications of the mentioned approaches in traffic management.

I did not detect flaws in the presentation of the approaches, procedures, methods, properties, features or citations of the references or their analysis.

I do not know relevant concepts, approaches, procedures or methods that are missing in the current proposal and that should be included.   

The included tables contains the definition of the acronyms and synthesis of the survey. They sustain the paper understanding.

Author Response

We would like to thank you for your acceptance.

Reviewer 3 Report

The paper "Intelligent traffic management in next-generation networks" is a survey of the state of art in using machine learning and deep learning for traffic classification, prediction and anomaly detection.

The related works section is sufficient and the references are novel. Many other papers are discussed throughout the paper in detail. Included are other survey papers that target the proposed aspects of traffic management. The paper outlines the contributions in chapter 1.2

Machine learning (with deep learning) is presented in chapter 2 and their use in SDNs is presented in chapter 3. The applications are summarized in tables that allow a good overview of the discussed subjects.

The use of the English language is good. The ideas are clearly presented.

The figures relevant and are easy to read.

It is a good paper that can be accepted for publishing.

Author Response

(The authors gave the same response as above.)
